# Development of a Highly Specific Fluoroimmunoassay for the Detection of Doxycycline Residues in Water Environmental and Animal Tissue Samples

**DOI:** 10.3390/mi13111864

**Published:** 2022-10-30

**Authors:** Tao Le, Rongli Xu, Lulan Yang, Yong Xie

**Affiliations:** 1College of Life Sciences, Chongqing Normal University, Chongqing 401331, China; 2Fuxing Hospital, Capital Medical University, Chongqing 401331, China; 3Bioassay 3D Reconstruction Laboratory, Chongqing College of Electronic Engineering, Chongqing 401331, China

**Keywords:** doxycycline, monoclonal antibody, quantum dot, fluoroimmunoassay

## Abstract

Doxycycline (DOX) and its metabolite residues in food and the environment pose a serious threat to human health and the ecological environment. In this work, a novel method, termed competitive fluoroimmunoassays (cFIA), based on monoclonal antibody (mAb) bio-conjugated CdSe/ZnS core–shell quantum dots (QDs), was developed for sensitive and rapid bioanalyses of DOX in natural water and commercial meats. After the optimization of the experimental conditions, 1 μg mL^−1^ of coating antigen and 0.5 μg mL^−1^ of QD-labeled mAb were used for the establishment of the cFIA. With this assay, the 50% inhibition concentration was found to be 0.35 ng mL^−1^ of DOX in phosphate-buffered saline samples, and the limit of detection was 0.039 ng mL^−1^ with minor cross-reactivity to other tetracycline members. The recoveries from natural water and commercial meats spiked with DOX concentrations of 10–600 ng mL^−1^ were 81.3–109.8%, and standard deviation were all below 12%. Levels measured with the QD-cFIA for thirty authentic samples were confirmed by high-performance liquid chromatography with good correlations. These results indicate that QD-cFIA is sultable for the rapid and quantitative detection of DOX residue in environmental and food samples.

## 1. Introduction

Doxycycline (DOX) is an essential member of the tetracycline family of antibiotics that is often used to treat infectious diseases in animals [1]. It is also used as a feed additive to prevent diseases and promote the growth of animals. Early research on DOX residues focused on food or biological samples [2]. Since animal manure and urine are usually used as fertilizers on farmland, more attention has been paid to the impact of residual DOXs on water and soil environments [3]. Depending on the animal species, researchers have found that up to 70% of DOX in urine or feces enters soil and surface water through the excretion of prototypes or metabolites.

According to different animal species, researchers have found that up to 70% of DOX in urine or feces enters the soil and surface water via excreta in prototypes or metabolites [4,5]. Notably, extensive use of animal fertilizers (not harmlessly treated) causes DOX to enter the soil and groundwater, thereby seriously threatening the ecological environment and the safety of agricultural products [6]. The presence of these antibiotic residues in the environment or food or biological samples can be directly toxic or can cause allergic reactions in some hypersensitive individuals, also promoting drug resistance to the microbial strain. Since these DOX residues are present in the environment or in food or biological samples, they may be directly toxic and cause allergic reactions in certain allergic individuals and can also promote resistance to microbial strains [7,8]. In addition, due to its broad-spectrum antimicrobial activity, residues in environmental media may affect the microbial abundance, community structure, and function, thus affecting the stability of the entire ecosystem [5,9].

Traditionally, the residues of DOX are measured by analytical methods. Electrochemistry [10], high-performance liquid chromatography tandem mass spectrometry [11], high-performance thin-layer chromatography [12]. and high-performance liquid chromatography (HPLC) [13,14] have been successfully used to detect DOX. However, these methods have some obvious defects, such as the need for expensive equipment and highly skilled personnel and may not be suitable for rapid screening of many samples. Additionally, immunoassays are widely used for the detection of various small molecule drugs because of their simplicity, rapidity, low cost, and high throughput. In our previous study, enzyme-linked immunosorbent assay [2,15], immunochromatographic test strip [16] and time-resolved fluroimmunoassay [17] have been developed to detect DOX in animal edible tissues (Table 1). However, these immunoassay methods based on horseradish peroxidase, colloidal gold, europium and samarium labelled antibodies to generate colored molecules as signal output have low sensitivity when detecting some trace antibiotics or biomarkers in complex biological matrices.

Recently, compared with traditional immunoassays, the use of fluoroimmunoassays (FIA) has been increasing due to their higher sensitivity, wider detection range, and lower matrix interference [18,19]. The fluorescent label may be an inorganic or organic material, such as CdSe/ZnS core–shell quantum dots (QDs), lanthanide chelates, fluorescent proteins, fluorescent microspheres, and so on [20,21]. As a class of fluorescent semiconductor nanocrystals with unique optical properties, such as a high quantum yield, high photostability, large molar extinction coefficient, wide absorption, and narrow fluorescence emission spectrum, QDs are considered to be promising fluorescent labeling materials for use in immunoassays [22]. At present, QDs have been widely used in the biomedical and food safety fields as labeling materials for immunofluorescence probes [22,23,24]. However, as yet, no FIA has been reported for DOX, and a simple and rapid QD-based FIA (QD-cFIA) is urgently needed.

In this work, the aim was to develop QD-cFIA for DOX in edible animal tissues and environmental samples using QD-labeled monoclonal antibodies (mAbs). The optimization of parameters, including the coating antigen concentration, QD-labeled mAb concentration, blocking buffer, incubation temperature, and competitive reaction time, were detailed. Finally, the proposed QD-cFIA was compared with conventional HPLC results in an analysis of environment and food samples containing DOX.

## 2. Materials and Methods

### 2.1. Materials and Instruments

Doxycycline (DOX), oxytetracycline, tetracycline, chlortetracycline, ovalbumin (OVA), and carboxylic acid-modified CdSe/ZnS core–shell QDs were purchased from Sigma–Aldrich (St. Louis, MO, USA). The coating antigens DOX-para-aminobenzoic acid–DOX (DOX-PABA-OVA) and anti-DOX mAb were obtained from our laboratory [2]. HPLC-grade acetonitrile and methanol were purchased from Fisher Chemical Company (Fair Lawn, NJ, USA). Ultraviolet absorbance was detected by using a NanoDrop-1000 spectrophotometer (Thermo Fisher Scientific Inc., Wilmington, DE, USA). The fluorescence was determined with a Varioskan LUX Multimode Microplate Reader (Thermo Fisher Scientific Inc., Wilmington, DE, USA). Deionized water was purified using the Millipore Milli-Q Ultrapure Water System (Bedford, MA, USA) and was used for all experiments. The QD-cFIA was validated with an Agilent 1260 HPLC equipped with an ultraviolet detector (Agilent, Wilmington, DE, USA).

### 2.2. Preparation of the Labeled Antibodies

The anti-DOX mAbs were covalently conjugated with the QDs by using carbodiimide chemistry. Briefly, 50 μL of QD was activated by adding 50 μL of 1-ethyl-3-(3-dimethylaminopropyl)carbodiimide hydrochloride (4 mg mL^−1^) and 50 μL of N-Hydroxysuccinimide (0.15 mg mL^−1^), and the mixture was incubated for 30 min at room temperature with gentle shaking. Then, the solution’s pH was adjusted to 7.4, and 300 μL of mAb (6 mg mL^−1^) was added to the ester-activated QD and then incubated for 2 h under dark conditions at room temperature. The reaction was separated by centrifugation at 18,000 rpm for 30 min at 4 °C, and the precipitates were dispersed in 0.05 M borate buffer (pH 8.0) until further use.

### 2.3. QD-cFIA Procedure Used for DOX Detection

As shown in Figure 1, the QD-cFIA procedure was carried out as follows: (1) The coating antigen (DOX-PABA-OVA) was diluted to a certain concentration with 0.05 M carbonate-buffered saline buffer (pH 9.6, 100 μL per well) and then added to a 96-well microplate. (2) After incubation overnight at 37 °C, the microplates were washed five times with 0.05 M sodium borate buffer (pH 9.0) containing 0.05% Tween-20 and then blocked with 1% OVA (200 μL per well) for 0.5 h at 37 °C. (3) After washing three times, 50 μL of the sample or standard solution and 50 μL of QD-labeled mAb were added to each well and incubated for 1 h at 37 °C. (4) The fluorescence intensity (F) of each well was measured with a Microplate Reader after washing and padding dry.

### 2.4. Optimization of Experimental Conditions

The experimental parameters were optimized in order to improve the sensitivity of the QD-cFIA. Coating antigens were diluted to concentrations ranging from 2 to 0.125 μg mL^−1^ by serial dilution with 0.05 M carbonate-buffered saline buffer (pH 9.6). The QD-labeled mAb (0.5 mg mL^−1^) was diluted with sodium borate buffer to concentrations of 4, 2, 1, 0.5, and 0.25 μg mL^−1^, respectively. The blocking buffers were selected, including 1% S-Milk, 1% BSA, 1% Gelatin, 1% OVA, and 1% Casein. The incubation temperatures used for the competitive binding assays were 4 °C, 25 °C, 37 °C, and 45 °C, respectively. The competitive reaction times were 15, 30, 45, 60, and 75 min, respectively. The concentration of DOX was diluted from 1000 to 0.01 ng mL^−1^ in sequence. F_0_/IC_50_ was used as the main index to evaluate the performance of QD-cFIA (F_0_: the fluorescence signal absence of analytes; IC_50_ represents the half-maximal inhibition concentration), where the highest F_0_/IC_50_ ratio was the most desirable [25].

### 2.5. Standard Curve and Specificity of QD-cFIA

To establish a standard curve, a series of DOX standard solutions were prepared by diluting DOX standards in PBS at a concentration of 0.01–100 ng mL^−1^. The standard curves for DOX were obtained by plotting the mean F/F_0_ against the logarithm of the analyte concentration under optimized parameters (F_0_ and F represent the fluorescence intensities without and with the analyte, respectively). Origin Pro 7.0 software was used to analyze the IC_50_ and limit of detection (LOD, IC_10_) using a four-parameter logistic equation of the Sigmoidal curve [26]. The specificity of QD-cFIA was evaluated by cross reactivity (CR). The CR values were calculated as follows: CR% = (IC_50_ of analyte/IC_50_ of analog) × 100 [2].

### 2.6. Accuracy and Precision Studies

The accuracy and precision of QD-cFIA were evaluated by the recovery rate and relative standard deviation (RSD). Animal tissue and water samples that had been shown to be DOX-free were used for matrix effect, accuracy, and precision studies. According to the Chinese MRLs standard, different concentrations of DOX were spiked to edible animal tissues, which were 0.5, 1 and 2 times the MRLs, respectively. Liver samples were spiked with DOX at final concentrations of 150, 300, and 600 μg kg^−1^ in sample buffer, and muscle and fish samples had final DOX concentrations of 50, 100, and 200 μg kg^−1^. All water samples were spiked with DOX at concentrations of 10, 50, and 100 μg L^−1^. DOX residues were extracted and purified from the samples using the modified QuEChERS (quick, easy, cheap, effective, rugged and safe) extraction method [27]. Each analysis was performed in quintuplicate.

### 2.7. The Correlation of QD-cFIA with HPLC

Thirty authentic samples of chicken liver, chicken muscle, fish muscle, tap water, river water, and contaminated water were collected from farms in Chongqing and Guizhou, China, where DOX had been used. All the samples were prepared using the procedure described above. Then, each sample was divided into two parts: one was analyzed using QD-cFIA and the other was analyzed using HPLC. For HPLC, the samples were analyzed in accordance with the method provided in the national standards of the Ministry of Agriculture of the People’s Republic of China (No. 958–2–2007). HPLC was performed on an Agilent ZORBAX SB–C18 (150 mm × 4.6 mm I.D., 5 μm, New Jersey, USA) using a mixture of methanol/acetonitrile/oxalic acid (7:8:85, *v/v/v*) as the mobile phase at a flow rate of 1.0 mL/min. The operating temperature of the column was set to 35 °C while the wavelength of the UV detector was set to 355 nm, and the injection volume was 50 μL.

## 3. Results and Discussion

### 3.1. Characterization of the QD-Labeled Antibodies

The fluorescence spectra of the QD-labeled mAbs had a maximum emission wavelength of 520 nm (λex = 400 nm), the same as that of the QDs, indicating that the excellent optical properties of the QDs conjugated with the mAbs were still preserved. The fluorescence intensity of QD-labeled mAbs was slightly weaker than that of the QDs due to the loss of QDs and fluorescence quenching during the reaction. In addition, the following QD-cFIA results indicated that the QDs were successfully linked to the mAbs against DOX.

### 3.2. Optimization of QD-cFIA

The F_0_/IC_50_ ratio was used to estimate the optimal parameter, with the highest ratio corresponding to the optimal parameter for QD-cFIA. QD-cFIA showed the highest F_0_/IC_50_ ratios when the coating antigen and QD-labeled mAbs were concentrated at 1 μg mL^−1^ (Figure 2a) and 0.5 μg mL^−1^ (Figure 2b), respectively. In this work, the effect of blocking buffers on the QD-cFIA performance was also studied. As shown in Figure 2c, 1% BSA was used as the blocking buffer to obtain the best sensitivity for the assay. An optimal incubation temperature of 37 °C was chosen (Figure 2d) and 60 min was selected as the optimal incubation time for the competitive binding assay (Figure 2e). Therefore, 1 μg mL^−1^ of coating antigen, 0.5 μg mL^−1^ of QD-labeled mAb, 1% BSA, 37 °C, and 60 min were selected as the optimal parameters for QD-cFIA in the experiment.

### 3.3. Standard Curve and Specificity of QD-cFIA

Competitive curves with final DOX concentrations of 0.003, 0.01, 0.03, 0.1, 0.3, 1, 3, 10, 30, and 100 ng mL^−1^ were run in PBS under the optimized parameters (Figure 3a). A standard curve with good linearity (Figure 3b) was prepared by using DOX concentrations of 0.03, 0.1, 1, and 3 ng mL^−1^ in PBS. The standard curves that were based on the DOX solvent calibration showed good linearity with R^2^ values equal to 0.9935. The IC_50_, LOD, and linear range of the method were 0.35, 0.039, and 0.03–3 ng mL^−1^, respectively. As shown in Table 2, the QD-cFIA showed negligible CR with the analogues (CR < 0.01%) except for 4-epi-doxycycline, which showed a CR of 51.5%. These results are similar to those reported in previous studies using ic-ELISA [2].

### 3.4. Accuracy and Precision

The accuracy and precision values of QD-cFIA were determined by analyzing the recovery and RSD. As illustrated in Table 3, the recovery rate of spiked samples was in the range of 81.3–109.8%, and the RSD was 4.2–11.9%. The results indicate an excellent performance by the QD-cFIA during the detection of DOX residues in edible animal tissue and environmental samples.

### 3.5. Correction of Immunoassays and HPLC

The developed QD-CFIA method and the HPLC reference method were used to compare and analyze the natural DOX-contaminated food and environmental samples to evaluate the reliability of the QD-cFIA method. Using QD-cFIA, we found that the samples were contaminated with various levels of DOX, ranging from 2.2 to 109.8 μg kg^−1^. The HPLC results were basically consistent with those of QD-cFIA, and the positive results ranged from 2.6 to 110.7 μg kg^−1^. The Bland–Altman plot for DOX measured by QD-cFIA and HPLC is shown in Figure 4. The results show that the mean bias was −3.6 μg kg^−1^, and the 95% limits of the agreement defined as the mean ± 1.96 SD were between –9.7 and 2.4 μg kg^−1^. It is evident that the results from the QD-cFIA and HPLC were not significantly different (significant level α = 0.05).

## 4. Conclusions

In this study, the proposed method was the first application of the QD-cFIA for the determination of DOX in edible animal tissue and environmental samples. The QD-cFIA showed high specificity for DOX and a qualitative LOD of 0.03 ng mL^−1^. Six samples were spiked with DOX and detected using the proposed QD-cFIA method, and an excellent recovery and satisfactory coefficient of variation were obtained. Additionally, the reliability of the developed QD-cFIA was confirmed by HPLC for parallel analysis of the natural DOX-contaminated food and environmental samples, further demonstrating that the established assay method can act as a sensitive and reliable tool for the detection of DOX residue in environmental and food samples.

## Figures and Tables

**Figure 1 micromachines-13-01864-f001:**
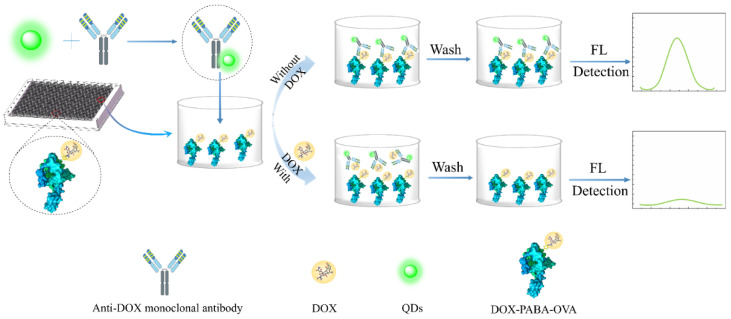
Schematic diagram of QD-cFIA.

**Figure 2 micromachines-13-01864-f002:**
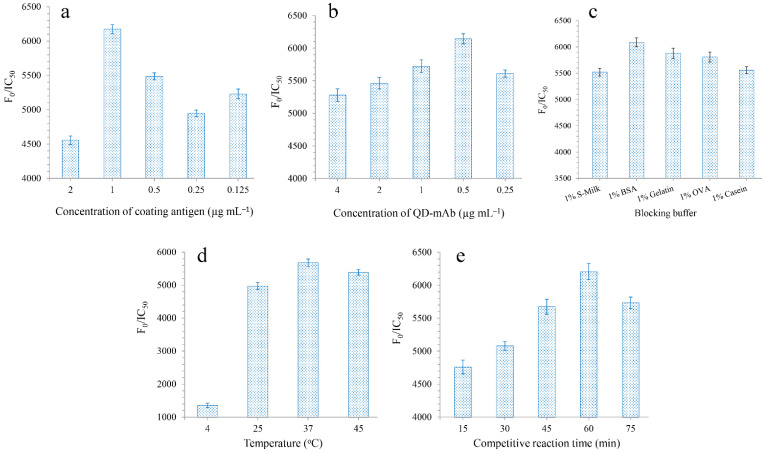
(**a**) Effect of DOX-PABA-OVA concentrations (2, 1, 0.5, 0.25 and 0.125 μg mL^−1^) on F_0_/IC_50_. (**b**) The ratio of F_0_/IC_50_ with different blocking buffer (1% S-Milk, 1% BSA, 1% Gelatin, 1% OVA and 1% Casein). (**c**) The ratio of F_0_/IC_50_ under different concentrations of QDs-labeled mAb (4, 2, 1, 0.5 and 0.25 μg mL^−1^). (**d**) The ratio of F_0_/IC_50_ under different incubation temperatures: 4 °C, 25 °C, 37 °C and 45 °C, respectively. (**e**) The ratio of F_0_/IC_50_ under different competitive reaction time: 15, 30, 45, 60 and 75 min, respectively. The error bars represent the standard deviation of three measurments.

**Figure 3 micromachines-13-01864-f003:**
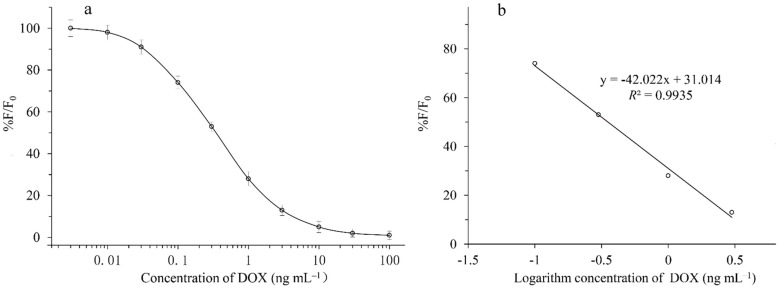
(**a**) Standard curves of our developed QD-cFIA with increasing DOX concentrations, from top to bottom: 0.003, 0.01, 0.03, 0.1, 0.3, 1, 3, 10, 30, and 100 ng mL^−1^, respectively. (**b**) Good linearity of the calibration curves was achieved for DOX in the range of 0.1–3 ng mL^−1^. Error bars represented the standard deviation.

**Figure 4 micromachines-13-01864-f004:**
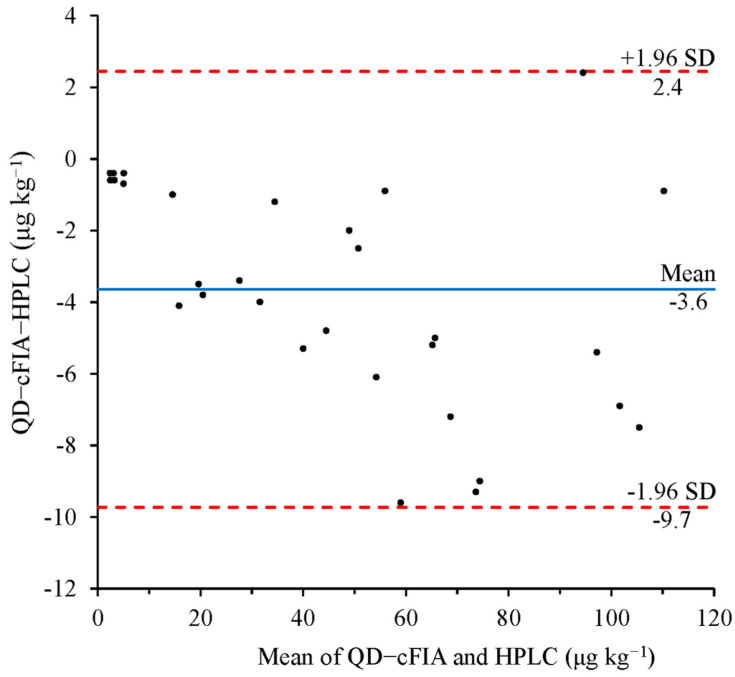
Bland–Altman plots to show comparisons between detection results obtained by the QD-cFIA and HPLC in fortified real samples.

**Table 1 micromachines-13-01864-t001:** A review of immunoassay methods for the determination of DOX in recent years.

Method	Antibody Type	IC_50_ (ng mL^−1^)	LOD (ng mL^−1^)	Sample	References
Enzyme-linked immunosorbent assay	Monoclonal antibody	1.32	0.14	liver, muscle and egg	[2]
Enzyme-linked immunosorbent assay	Polyclonal antibody	8.74	1.96	Liver and muscle	[15]
Immunochromatographic test strip	Polyclonal antibody	22.0	7.0	Liver and muscle	[16]
Time-resolved fluroimmunoassay	Polyclonal antibody	1.06	0.04	liver, muscle and egg	[17]
QD-cFIA	Monoclonal antibody	0.35	0.039	Chicken (liver, muscle), fish muscle, tap water, river, water and contaminated water	This work

**Table 2 micromachines-13-01864-t002:** Cross-reactivity of analogues related to DOX by QD-cFIA.

Compound	IC_50_ (ng m L^−1^)	CR (%)
DOX	0.35	100
4-epi-doxycycline	0.68	51.5
Oxytetracycline	>10,000	<0.01
4-epi- oxytetracycline	>10,000	<0.01
Tetracycline	>10,000	<0.01
4-epi-tetracycline	>10,000	<0.01
Chlortetracycline	>10,000	<0.01
4-epi-chlortetracycline chlortetracycline	>10,000	<0.01
Demeclocycline	>10,000	<0.01

**Table 3 micromachines-13-01864-t003:** Accuracy and precision of DOX in spiked samples by QD-cFIA (n = 5).

Sample	Spiked (ng mL^−1^, ng g^−1^)	Mean Recovery ±SD (%)	RSD (%)
Chicken liver	150	105.4 ± 10.4	9.9
	300	98.7 ± 10.8	10.9
	600	88.5 ± 9.7	11.0
Chicken muscle	50	98.8 ± 6.6	6.7
	100	109.8 ± 9.4	8.6
	200	108.4 ± 7.5	6.9
Fish muscle	100	81.3 ± 9.7	11.9
	200	91.8 ± 7.2	7.8
	400	109.3 ± 11.2	10.3
Tap water	10	102.3 ± 6.4	6.3
	50	103.1 ± 7.6	7.4
	100	98.8 ± 4.6	4.7
River water	10	103.3 ± 4.3	4.2
	50	101.5 ± 9.5	9.4
	100	99.6 ± 11.2	11.2
Contaminated water	10	106.3 ± 9.6	9.0
	50	100.8 ± 7.9	7.8
	100	97.3 ± 9.5	9.8

## Data Availability

The data presented in this study are available on request from the corresponding author.

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
