# Peer review of "Development of a Highly Specific Fluoroimmunoassay for the Detection of Doxycycline Residues in Water Environmental and Animal Tissue Samples"

_micromachines, 2022, doi:10.3390/mi13111864_

Round 1
Reviewer 1 Report
In this paper, the authors developed a novel competitive fluoroimmunoassays (cFIA) based on anti-DOX monoclonal antibody (mAb) bio-conjugated CdSe/ZnS core-shell quantum dots (QDs) for sensitive and rapid bioanalyses doxycycline (DOX) in natural water and commercial meats. Good analytical performance and results were obtained due to the introduction of CdSe/ZnS core-shell quantum dots (QDs) in this cFIA. After some necessary revisions, I recommend this paper to publish in the Micromachines.
1. I suggest that the authors could find a native speaker or editorial company to revise the paper especially the language, grammar and some specialized vocabulary.
2. In the part of Introduction, the authors should add the advantages of CdSe/ZnS core-shell quantum dots (QDs) used in this cFIA.
3. There is no description about the pretreatment of real samples, please add the corresponding part.
4. In the Results and discussion part, there is no characterization about the structure and fluorescence property of CdSe/ZnS core-shell quantum dots (QDs) which is the important content for the integrality of a research.
5. The full names of some specialized vocabulary were required when they appeared in the paper firstly.
6. The format of Table 2 should be revised.
7. I suggest the author to rewrite the conclusion part to summary the main content of this work better.
8. The reference format should be revised according to the latest papers published in the Micromachines.
Author Response
We deeply appreciate your constructive and positive comments for improvements regarding to our manuscript. Our work haves benefited substantially from your invaluable input. Thank you very much.
- I suggest that the authors could find a native speaker or editorial company to revise the paper especially the language, grammar and some specialized vocabulary.
Response: Thank you for your constructive suggestion. We have asked the editorial company to make changes to the language, grammar and some specialized vocabulary of the revised manuscript, see line revised manuscript and English-Editing-Certificate-52243.
- In the part of Introduction, the authors should add the advantages of CdSe/ZnS core-shell quantum dots (QDs) used in this cFIA.
Response: Thank you very much for your invaluable comments and suggestions. We have corrected “Introduction” and described the advantages of CdSe/ZnS core-shell quantum dots (QDs) in the text, see lines 67-72.
- There is no description about the pretreatment of real samples, please add the corresponding part.
Response: Thank you very much for your invaluable comments and suggestions. We have writed more detailed information on sample pretreatment on parts “2.6. Accuracy and precision studies” to “DOX residues were extracted and purified from the samples using the modified QuEChERS (quick, easy, cheap, effective, rugged and safe) extraction method [24]”, see lines 148-150.
- In the Results and discussion part, there is no characterization about the structure and fluorescence property of CdSe/ZnS core-shell quantum dots (QDs) which is the important content for the integrality of a research.
Response: Thank you for your suggestion. We have characterized QD in the Results and Discussion, only described it with language “The fluorescence spectra of the QDs-labeled mAbs also had a maximum emission wavelength of 520 nm (λex = 400 nm),”, without showing the representation Figure. Because the characterization of QD is very simple and has been reported in many literatures, it is not necessary to add another graph separately, see lines 66-67.
- The full names of some specialized vocabulary were required when they appeared in the paper firstly.
Response: We have checked the abbreviations of the manuscript, and the first abbreviations show their full names.
- The format of Table 2 should be revised.
Response: We have corrected it, see Table 2.
- I suggest the author to rewrite the conclusion part to summary the main content of this work better.
Response: Thank you for your suggestion.We have corrected the conclusion part, see Conclusions and lines 230-238.
- The reference format should be revised according to the latest papers published in the Micromachines.
Response: Thank you for your suggestion. We have modified the format of the reference as required by Micromachines.
Reviewer 2 Report
The authors developed a quantum dots (QDs) based competitive fluoroimmunoassay method was developed for sensitive and rapid bioanalyses of DOX in water and animal tissue (liver and muscles) samples using anti-DOX monoclonal antibody bio-conjugated CdSe/ZnS core-shell QDs. A competitive fluorescent QD-immunoassay is not a novel technique, and it was employed in many papers for screening of many antibiotic residues in animal tissues. However, this application could be suitable for publication in micromachines after addressing the following isssues:
1. The authors are asked to compare between the developed method and the previously reported methods for DOX immunosensing in terms of sensitivity, selectivity, practicability, rapidity, and stability to illustrate why this developed method is superior to the previously developed ones. This comparison could be performed in a tabular form.
2. Title: “fluorimmunoassay” must be corrected to be “fluoroimmunoassay”. Also, it is better to be “environmental water” instead of “water environmental”.
3. The manuscript should also be revised carefully for language and grammatical errors because it contains some linguistic and grammatical mistakes that require correction.
Author Response
Revision on comments of referee 2
We deeply appreciate your constructive and positive comments for improvements regarding to our manuscript. Our work have benefited substantially from your invaluable input. Thank you very much.
1.The authors are asked to compare between the developed method and the previously reported methods for DOX immunosensing in terms of sensitivity, selectivity, practicability, rapidity, and stability to illustrate why this developed method is superior to the previously developed ones. This comparison could be performed in a tabular form.
Response: Thank you for your kind advice, we have compared the detection performance of the established method with previously reported DOX immunoassay methods, see Table 1.
- Title: “fluorimmunoassay” must be corrected to be “fluoroimmunoassay”. Also, it is better to be “environmental water” instead of “water environmental”.
Response: It has been corrected in the revised MS, see line 2 and line 34.
- The manuscript should also be revised carefully for language and grammatical errors because it contains some linguistic and grammatical mistakes that require correction.
Response: Thank you for your constructive suggestion. We have corrected the language and grammar errors in revised manuscript.
Round 2
Reviewer 1 Report
We think that the manuscript could be accepted at the current form.
Author Response
Response to Reviewer 1 Comments
Point 1: We think that the manuscript could be accepted at the current form.
Response 1: Thank you for your review comments.